# AdvMem: Adversarial Memory Initialization for Realistic Test-Time Adaptation via Tracklet-Based Benchmarking

## Abstract

We introduce a novel tracklet-based dataset, Inherent Temporal Dependencies (ITD), to benchmark test-time adaptation (TTA) methods under realistic temporal dependencies. Unlike existing benchmarks that primarily study distribution shifts and violations of the i.i.d. assumption, ITD captures sequences of object-centric images (tracklets) from object-tracking datasets, reflecting the temporal correlations seen in real-world video streams. Using this dataset, we analyze current TTA methods and highlight their limitations under temporally correlated data. Building on these insights, we propose an adversarial memory initialization strategy that significantly improves the performance of memory-based TTA methods on our challenging benchmark.

## 1 Introduction

Deep neural networks (DNNs) have demonstrated impressive performance across various domains He et al. (2016). However, their reliability often diminishes in real-world scenarios due to natural corruptions and distribution shifts Hendrycks et al. (2021); Hendrycks & Dietterich (2019); Kar et al. (2022). These shifts can manifest as unforeseen distortions that cause the input data to deviate from the model's training distribution. Additionally, the distribution of image classes may differ from what the model has learned, further compounding the challenge. Consider images captured by hand-held cameras—these introduce two major difficulties: (1) the visual distribution may differ significantly from training data, such as in foggy or rainy conditions, and (2) images arrive sequentially as part of a continuous video stream. The first issue represents a distribution shift in the visual space, while the second introduces temporal dependencies that break the *independence* assumption inherent in conventional training (i.i.d.). Addressing both aspects is crucial for ensuring the robustness of DNNs in practical deployments.

**Test-Time Adaptation and its Challenges.** Test-Time Adaptation (TTA) seeks to mitigate performance degradation by adapting a pre-trained model on-the-fly using an incoming data stream Liang et al. (2023). At inference, TTA methods perform online unsupervised learning to adjust model parameters in response to new data Liang et al. (2020a); Sun et al. (2020a); Wang et al. (2020b); Iwasawa & Matsuo (2021). While TTA has shown promise, current benchmarks oversimplify the problem, primarily simulating distribution shifts without accounting for temporal dependencies that violate the i.i.d. assumption. For instance, many TTA approaches Liang et al. (2020a) assume that distribution shifts are purely covariate shifts Candela et al. (2009), as seen in datasets like CIFAR10-C and ImageNet-C Hendrycks & Dietterich (2019) (Fig. 1, left). Meanwhile, other methods Boudiaf et al. (2022) address non-i.i.d. scenarios by modifying label distributions while neglecting the visual continuity inherent in sequential data. A more recent effort by Yuan *et al.* Yuan et al. (2023) considers both distribution shifts and non-i.i.d. labels, but still overlooks the critical role of temporal dependencies. We argue that the lack of benchmarks that jointly capture distribution shifts and temporal dependencies has limited the development of deployable TTA methods. To bridge this gap, we take inspiration from object tracking to introduce a benchmark that inherently accounts for both challenges.

**Introducing ITD.** Our benchmark, Inherent Temporal Dependencies (ITD), is built using tracklets—short sequences of images tracking the same object across consecutive frames. By leveraging

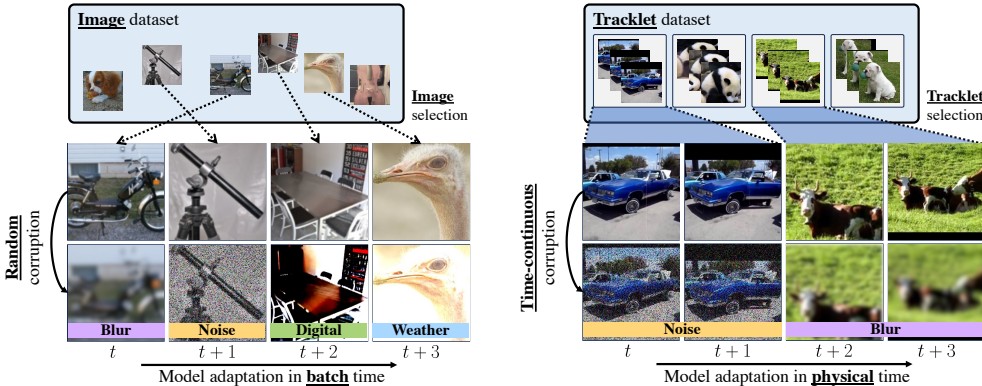

Figure 1: **A tracklet-based benchmark for realistic evaluation of Test-Time Adaptation (TTA) methods (Inherent Temporal Dependencies).** *(Left)* Existing benchmarks evaluate TTA methods using streams of images depicting different objects across batches, with random corruptions applied independently to each image. *(Right)* Our proposed ITD benchmark addresses these limitations by (i) presenting images of the same object in a sequence, preserving temporal dependencies from real-world tracklets, and (ii) applying consistent corruptions whose intensity may evolve over time. This framework offers a more realistic setting for evaluating the adaptability of TTA methods.

TrackingNet Muller et al. (2018), we create a realistic test-time adaptation setting where temporal dependencies naturally emerge, leading to i.i.d. violations. To introduce controlled distribution shifts, we apply standard transformations and corruptions Hendrycks & Dietterich (2019); Kar et al. (2022) (*e.g.* Gaussian noise, glass blur) consistently across tracklets rather than as independent perturbations (Fig. 1, right). This setup better reflects real-world challenges by aligning the temporal structure of the dataset with the sequential nature of data streams, following a protocol inspired by RoTTA Yuan et al. (2023). Using ITD, we rigorously analyze how temporal dependencies interact with distribution shifts, revealing significant weaknesses in existing TTA methods.

**Advancing TTA with ADVMEM.** Our investigation highlights that most TTA methods struggle under the compounded effects of distribution shifts and temporal dependencies, leading to severe performance drops. We examine memory-bank-based methods, which should, in principle, handle temporal challenges well due to their ability to adapt to selectively stored samples. However, our results show that these methods suffer from poor initialization of the memory bank, significantly impacting performance.

To address this, we propose ADVMEM, a novel adversarial memory initialization strategy that enhances stability in adaptation. By leveraging synthetic noise generated in a class-diverse manner, ADVMEM operates as a plug-and-play enhancement for memory-based TTA methods. Experiments demonstrate its effectiveness—equipping SHOT-IM Liang et al. (2020b), with ADVMEM reduces error rates by 44% in Tracklet-Wise i.i.d. settings (see Table 2).

**Our Contributions.**

- **ITD Benchmark.** We introduce Inherent Temporal Dependencies, a novel benchmark for TTA that integrates object tracklets, capturing real-world distribution shifts and temporal dependencies.
- **Comprehensive Evaluation of TTA Methods.** We systematically assess existing TTA approaches on ITD, highlighting their limitations under realistic non-i.i.d. conditions.
- **ADVMEM.** We equip existing TTA methods with memory and benchmark their memory-adapted versions, demonstrating the impact of incorporating memory mechanisms on adaptation performance. Additionally, we propose an adversarial memory initialization strategy that significantly improves the performance, particularly under severe non-i.i.d. scenarios.

Our work advances the study of test-time adaptation by providing a more realistic evaluation framework and a novel solution to enhance the stability of model adaptation in dynamic environments.

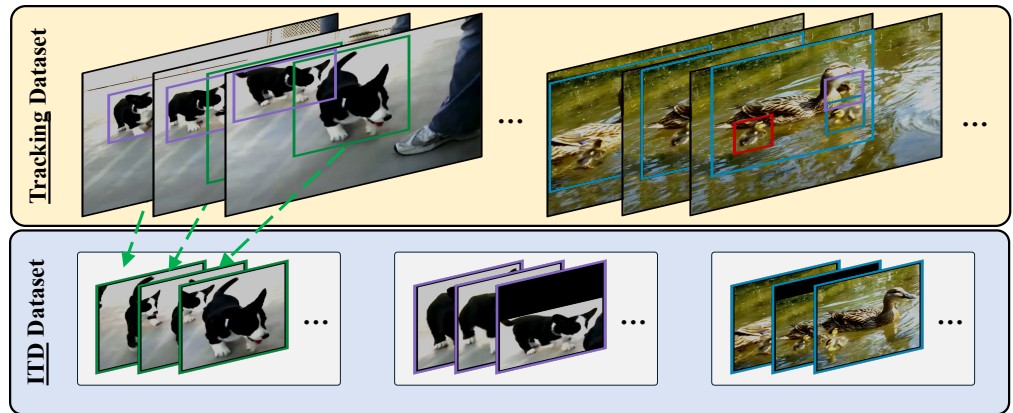

Figure 2: **We build ITD with realistic TTA instances by constructing them from a tracking dataset.** We extract object-centric sequential video frames, encapsulating the small variations of the same entity over time. We source the frames and bounding boxes from TrackingNet, a well-established tracking dataset, such that the instances focus on particular objects of interest. As such, these sequences naturally exhibit the temporal dependencies inherent in real-world scenarios.

## 2 RELATED WORK

**Test-Time Adaptation.** TTA leverages the unlabeled data that arrives at test time to adapt the forward pass of pre-trained DNNs according to some proxy task Wang et al. (2020a); Liang et al. (2020b). Many existing TTA methods focus on covariate distribution shifts Li et al. (2016); Niu14 et al.; Niu et al. (2022); Wang et al. (2022); Liang et al. (2020b). Several TTA methods tackle this challenge by updating the statistics of the Batch Normalization layers at test time Li et al. (2016); Schneider et al. (2020). For example, AdaBN Li et al. (2016) introduces Adaptive Batch Normalization, an algorithm to adapt to the target domain. Another group of methods uses an entropy minimization strategy. For instance, TENT Wang et al. (2020a) minimizes the entropy of the model's predictions. ETA and EATA Niu et al. (2022) extend TENT by selecting reliable and non-redundant samples to update the model weights. More recently, RoTTA Yuan et al. (2023) attempts to combat non-i.i.d. streams at test time by leveraging a memory bank for adapting to an incoming stream of data. In this work, we introduce ADVMEM, a novel adversarial memory initialization strategy to significantly enhance the adaptability of TTA methods under complex, non-i.i.d. scenarios.

**Benchmarking TTA Methods.** The fundamental premise of TTA involves deploying a pre-trained model onto edge devices like self-driving cars or surveillance cameras, where it faces potential changes in data distribution Liang et al. (2020a); Sun et al. (2020b); Wang et al. (2020b); Iwasawa & Matsuo (2021). This scenario unfolds as the model encounters a continuous stream of data, with each input potentially coming from a distribution different than the one the model was originally trained on. To emulate such scenarios, the TTA literature commonly creates a stream of data with samples from the test set of well-established image classification datasets, such as ImageNet Deng et al. (2009) and CIFAR Krizhevsky et al. (2009). Setups then systematically simulate covariate distribution shifts by inducing corruptions on individual images, such as those from Common Corruptions Hendrycks & Dietterich (2019) and 3D Common Corruptions Kar et al. (2022). In this work, we present a comprehensive benchmark for simulating more realistic and complex scenarios.

## 3 DATASET AND METHODOLOGY

**Motivation.** Recent advances in Test Time Adaptation (TTA) have moved beyond traditional i.i.d. setups toward more realistic non-i.i.d. configurations. RoTTA Yuan et al. (2023) introduced correlation sampling to model label dependencies observed in practice (e.g., frequent "pedestrian" labels in crowded scenes), revealing that existing TTA methods struggle under such streams.

In real-world data (e.g., surveillance videos), consecutive frames often show the same object with minor variations, creating strong temporal dependencies. To illustrate their impact, we ran a simple

CIFAR-10-C experiment: comparing PTTA Yuan et al. (2023) evaluation against a modified setting where each batch duplicates images to mimic tracklets. As shown in Table 1, all methods degrade notably under this tracklet mimic, underscoring the challenge posed by temporal redundancy.

These insights motivate our ITD benchmark, designed to capture both label and visual non-i.i.d. shifts, and our ADVMEM plugin, which mitigates over-adaptation under such conditions.

**Tracklets: A Natural Source of Visual Non-i.i.d. Data.** To construct a realistic non-i.i.d. benchmark, we leverage the field of Object Tracking. Unlike artificially simulated correlation sampling used in prior works Yuan et al. (2023), object-tracking datasets inherently capture realistic temporal dependencies by tracking specific objects across video frames. We propose using *tracklets*—sequences of object-centric images extracted from tracking datasets to model the gradual variations encountered in real-world image streams. This approach not only enhances the realism of our benchmark but also faithfully captures intrinsic characteristics of natural image sequences, establishing a robust foundation for evaluating TTA in real-world scenarios.

Table 1: **Average Error Rates on CIFAR-10-C.** Comparison of non-i.i.d. episodic vs. tracklet-mimic evaluation (averaged across corruptions), where the mimic setting simulates real-world temporal dependencies.

| Method | Non i.i.d. (%) | Mimic (%) | Δ (%) |
|---|---|---|---|
| Source | 44.1 | **44.1** | **0** |
| AdaBN | 75.4 | 78.7 | 3.3 |
| CoTTA | 75.5 | 89.1 | 13.6 |
| SAR | 75.2 | 82.4 | 7.2 |
| ETA | 75.4 | 78.6 | 3.2 |
| TENT | 75.3 | 85.1 | 9.8 |
| RoTTA | 27.6 | 66.8 | 39.2 |

### 3.1 DATASET CONSTRUCTION

We build Inherent Temporal Dependencies (ITD) from the large-scale TrackingNet Muller et al. (2018), which itself is derived from YouTube Bounding-Boxes Real et al. (2017).

For each video, we extract *tracklets* by following an object across frames and cropping its bounding boxes, yielding sequences that capture realistic temporal variations (Figure 2). Preprocessing steps include sampling every 5th frame to balance dataset size and redundancy, enlarging bounding boxes by 10% before cropping to preserve context, and resizing all crops to $224{\times}224$ for consistency with batch-based training.

**Dataset Properties.** Unlike conventional datasets where samples are independent, ITD is composed of *tracklets*, preserving the temporal continuity of objects. Each tracklet consists of images depicting the same object in different frames, naturally encoding non-i.i.d. dependencies. Additionally, our dataset supports temporally consistent corruptions (detailed in Section 4.4), further enhancing its relevance for evaluating TTA under realistic conditions.

**Statistics.** The ITD dataset contains over 23K objects that span 21 classes. The dataset is divided into training (50%), validation (30%), and test (20%) sets. In total, it comprises over 220K images—more than four times the size of ImageNet-C (50K)—while also providing object-instance relationships via tracklets. Further statistics, including class distributions, are provided in the appendix underscoring the scale and diversity of ITD, making it a valuable resource for advancing TTA research.

## 4 BENCHMARKING ON ITD

**Overview.** Unlike previous benchmarks that assume independent samples, ITD introduces a tracklet-based evaluation to reflect real-world challenges, such as sequential dependencies and non-i.i.d. distributions. We systematically evaluate TTA methods across three levels of complexity to assess their adaptability in streaming environments:

- **Frame-wise i.i.d. (Section 4.5):** Frames are sampled independently and identically distributed (i.i.d.), without considering sequential dependencies.
- **Tracklet-wise i.i.d. (Section 4.6):** Entire tracklets are sampled i.i.d., preserving intra-tracklet dependencies while maintaining inter-tracklet randomness.

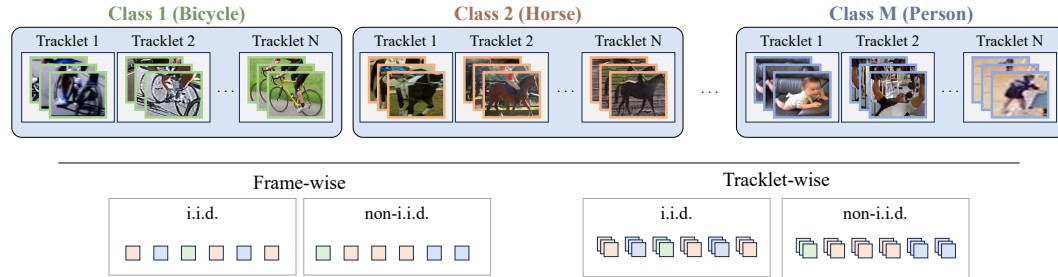

Figure 3: **Frame-wise and Tracklet-wise Experiment Setup:** We illustrate the construction of the frame-wise and tracklet-wise experiments. In the frame-wise setup, one frame is sampled from each tracklet to ensure each object is observed once. In the tracklet-wise setup, the frames within each tracklet are sequentially processed. Both i.i.d. and non-i.i.d. settings are depicted for each setup.

- **Tracklet-wise non-i.i.d. (Section 4.7):** Tracklets are sampled following a Dirichlet distribution Yuan et al. (2023), enforcing stronger non-i.i.d. properties across the dataset.

These setups enable a systematic evaluation of TTA methods under increasingly complex real-world conditions. Each scenario is tested under a single domain shift (e.g., fog) to isolate the effect of adaptation techniques.

## 4.1 TRACKLET-BASED ADAPTATION STRATEGIES

Unlike frame-wise adaptation, tracklet-based setups introduce sequential dependencies, making naive entropy minimization unreliable due to biased batch statistics. To ensure fair comparisons, we extend TENT and SHOT-IM by incorporating RoTTA's memory bank $\mathcal{M}$. This allows them to store and utilize previously seen samples, ensuring more stable adaptation in non-i.i.d. streams. In these adaptations, we replace RoTTA's original objective with entropy minimization for TENT and information maximization for SHOT-IM.

## 4.2 EXPERIMENTAL SETUP

Having established the dataset structure, corruption strategies, and adaptation mechanisms, we now outline our standardized experimental setup for evaluating TTA methods. Unless stated otherwise, we use ResNet-18 as the base model, apply corruptions at the highest severity level (5), and set a batch size of 64 for streaming data. We evaluate eight TTA methods and compare them against a pre-trained model $f_\theta$(Source) as the baseline. The details of these methods are provided in Table 2. Each method is assessed using its optimal hyperparameters, determined through an extensive search.

Table 2: Overview of TTA Methods Evaluated in ITD.

| Method | Adaptation Strategy |
|---|---|
| **AdaBN** Li et al. (2016) | Updates batch normalization statistics. |
| **SHOT-IM** Liang et al. (2020b) | Maximizes mutual information. |
| **TENT** Wang et al. (2020a) | Utilizes entropy minimization for adaptation. |
| **SAR** Niu14 et al. | Employs sharpness-aware optimization. |
| **EATA** Niu et al. (2022) | Entropy-based sample selection. |
| **CoTTA** Wang et al. (2022) | Uses consistency-based distillation for continual adaptation. |
| **RoTTA** Yuan et al. (2023) | Maintains a memory bank to stabilize adaptation. |

## 4.3 CONSIDERED CORRUPTIONS

To evaluate robustness, we apply distribution shifts from ImageNet-C, covering **Noise** (Gaussian, Shot, Impulse), **Blur** (Defocus, Glass, Zoom), **Weather** (Snow, Frost, Fog, Brightness), and **Digital Artifacts** (Contrast, Elastic Transform, Pixelate, JPEG Compression).

Table 4: **Performance Comparison under Frame-wise i.i.d. Assumption.** Average error rates reported for TTA methods on images from the ITD dataset. In this setup, all images/frames are shuffled and then independently subjected to corruptions. Notably, SHOT-IM outperforms other methods across all corruptions, showcasing its robustness against these domain shifts.

| Method | Source | AdaBN | SHOT-IM | TENT | SAR | CoTTA | ETA | EATA | RoTTA |
|---|---|---|---|---|---|---|---|---|---|
| Avg. Err. ↓ | 62.4 | 46.9 | **39.3** | 46.8 | 46.5 | 46.7 | 46.7 | 46.8 | 53.4 |

Given the temporal nature of ITD, we also explore scenarios where corruption severity varies within a tracklet, simulating transient environmental fluctuations (e.g., changing weather conditions). Additional extended evaluations can be found in the appendix.

### 4.4 PREPARATION FOR TTA

To evaluate the effectiveness of TTA methods on our ITD benchmark, we require models that are either pre-trained or fine-tuned specifically on this dataset. While ITD shares some class overlap with ImageNet, its distribution differs significantly, limiting the effectiveness of direct transfer. Our experiments confirm that ImageNet-pretrained models struggle to generalize well to ITD, highlighting the need for dataset-specific fine-tuning. Therefore, we fine-tune two ImageNet-pretrained models, ResNet-18 and ViT-B-16, on ITD's training set (see Table 3).

### 4.5 FRAME-WISE I.I.D. SCENARIO

In this scenario, we evaluate TTA methods under a conventional setting where each frame is treated as an independent and identically distributed (i.i.d.) sample, ignoring temporal dependencies. This setup assumes a uniform label distribution across time, thereby oversimplifying real-world conditions where label distributions may be highly imbalanced.

Table 3: **Error rates on all splits of ITD.** Detailed training and test loss/accuracy results are provided in the appendix.

| Error Rate | ResNet-18 | ViT |
|---|---|---|
| Train | 4.1 | 2.0 |
| Validation | 8.2 | 3.8 |
| Test | 9.4 | 4.0 |

To construct the test stream, we shuffle tracklets and sample one frame per tracklet, eliminating the notion of temporal continuity. Following standard practice Hendrycks & Dietterich (2019); Kar et al. (2022), we assess performance degradation under 15 different corruptions. Table 4 presents the average error rates of the eight evaluated TTA methods. As expected, distribution shifts significantly degrade the performance of pre-trained models. For example, the error rate of the Source model (ResNet-18 without TTA) rises from below 10% (Table 3) to over 60% (Table 4). Notably, TENT Wang et al. (2020a) reduces the error rate to below 50% through entropy minimization, while SHOT-IM achieves the lowest error, averaging below 40%.

### 4.6 TRACKLET-WISE I.I.D. SCENARIO

To move towards a more realistic evaluation, we introduce a setup where the model processes entire tracklets at test time. Within each tracklet, consecutive frames share labels and contextual consistency, though tracklets are sampled in an i.i.d. manner. This setup emulates real-world scenarios where the model observes a single object over multiple frames before switching to a new one.

We evaluate SHOT-IM (the best performer from Section 4.5), TENT, and RoTTA. Although RoTTA performed poorly in the frame-wise i.i.d. scenario, its memory-based approach is specifically designed for non-i.i.d. streams, making it relevant for this setting. To ensure a fair comparison, we extend TENT and SHOT-IM by incorporating RoTTA's memory bank, allowing them to only adapt to informative samples selected and retained in memory based on RoTTA Category-balanced sampling heuristics.

Table 5 summarizes our results. We find that SHOT-IM and TENT struggle under tracklet-based evaluation, with SHOT-IM's error rate increasing from under 40% (Table 4) to nearly 95% (Table 5).

This decline stems from the biased statistics computed within tracklets, which skew entropy-based adaptation. In contrast, RoTTA demonstrates superior stability, reducing the average error to around 50%, due to its distillation-based approach.

### 4.7 TRACKLET-WISE non-i.i.d. SCENARIO

In this final and most challenging setup, tracklets are sampled non-i.i.d. to simulate real-world streaming conditions such as autonomous driving or surveillance, where object categories appear in bursts. To model this, we follow Yuan *et al.* Yuan et al. (2023) and sample tracklets using a Dirichlet distribution $\text{Dir}(\gamma)$. As $\gamma \to 0$, label correlation within the stream increases, deviating from the i.i.d. assumption.

We evaluate RoTTA, SHOT-IM, and TENT under $\gamma = 10^{-4}$ to enforce strong non-i.i.d. conditions (additional $\gamma$ values are analyzed in Section 5). Table 5 reports the results. Even RoTTA, designed for non-i.i.d. streams, experiences a 28% performance drop compared to the tracklet-wise i.i.d. setup

Table 5: **Tracklet-wise i.i.d. vs. *non*-i.i.d. Evaluation with and without Memory.** Average error rates of TTA methods on ITD under i.i.d. (entire tracklets sampled independently) and *non*-i.i.d. (labels correlated via Dirichlet sampling) scenarios. Results are grouped by memory presence (✓) vs. absence (✗). RoTTA leverages memory to achieve strong robustness, clearly outperforming non-memory variants across corruptions and particularly excelling under *non*-i.i.d. streams.

| Method | Memory | Tracklet i.i.d. | Tracklet *non*-i.i.d. |
|--------|--------|-----------------|------------------------|
| TENT | ✗ | 94.0 | 94.0 |
|  | ✓ | 93.8 | 93.8 |
| SHOT-IM | ✗ | 94.7 | 95.1 |
|  | ✓ | 93.4 | 93.6 |
| RoTTA | ✓ | **51.3** | **79.3** |

(Table 5). We hypothesize that this decline results from imbalanced memory due to empty memory initialization, where certain classes are observed late in the stream, causing forgetting effects. These findings suggest that memory-based TTA methods can be further improved by introducing class-balancing initialization mechanisms to stabilize adaptation over long, non-i.i.d. streams

## 5 EXPERIMENTS

In Section 4.6, we observed that equipping TENT and SHOT-IM with a memory bank, while seemingly beneficial, does not yield significant performance improvements. Furthermore, in Section 4.7, we demonstrated how a tracklet-wise non-i.i.d. stream severely impacts RoTTA's performance. These findings indicate that while memory banks are essential for adapting to non-i.i.d. streams Yuan et al. (2023), even strong TTA methods experience significant degradation when evaluated on ITD.

We hypothesize that this degradation stems from the empty initialization of the memory bank. Consider the extreme case where $\gamma \to 0$, resulting in the memory bank lacking any examples for labels revealed later in the stream until those labels actually appear. Additionally, methods such as TENT rely on statistical measures like entropy for updates. When key classes are absent from the memory bank, model updates become skewed, leading to catastrophic forgetting. This motivates the need for a carefully designed memory initialization strategy that ensures stable adaptation steps.

### 5.1 ADVERSARIAL MEMORY INITIALIZATION

To address these challenges, we propose a novel memory bank initialization that stores class-representative samples, aiming to **(i)** prevent forgetting by covering all classes and **(ii)** balance output-space statistics.

To that end, we propose initializing the memory bank with synthetic data generated by adversarial algorithms Tzeng et al. (2017). Specifically, each memory bank entry is initialized as Gaussian noise, assigned a random label, and subjected to a targeted adversarial attack that maximizes the network's confidence in classifying it correctly, following Alfarra et al. (2022); Goodfellow et al. (2014). Formally, let $\mathcal{M}$ be an initially empty memory bank with a maximum capacity of $N$. We populate $\mathcal{M}$ iteratively with synthetic examples $x^*$, where:

$$x^* = \arg\min_x \mathcal{L}_{\text{ce}}(f_\theta(x), y), \tag{1}$$

Table 2: **Effect of Adversarial Memory Initialization on TTA in Tracklet-Wise i.i.d. Scenario.** On ITD, we compare standard (✗) and ADVMEM (✓) memory initialization. Equipping SHOT-IM with ADVMEM yields the best performance, highlighting its ability to enhance adaptability.

| Method | ADVMEM | Noise | | | Blur | | | Weather | | | | Digital | | | | Avg. |
|---|---|---|---|---|---|---|---|---|---|---|---|---|---|---|---|---|
| | | gauss. | shot | impul. | defoc. | glass | zoom | snow | frost | fog | brigh. | contr. | elast. | pixel. | jpeg | |
| TENT | ✗ | 94.2 | 94.2 | 94.3 | 94.5 | 94.5 | 93.9 | 93.7 | 93.7 | 93.2 | 93.0 | 92.8 | 93.8 | 93.4 | 93.5 | 93.8 |
| | ✓ | 85.1 | 83.2 | 88.2 | 82.9 | 84.8 | 74.0 | 85.5 | 84.2 | 83.9 | 68.9 | 71.6 | 79.9 | 62.6 | 69.5 | 78.9 |
| SHOT-IM | ✗ | 93.8 | 93.9 | 93.9 | 94.2 | 94.3 | 93.7 | 93.3 | 93.0 | 93.0 | 92.8 | 92.1 | 93.5 | 93.4 | 93.2 | 93.4 |
| | ✓ | **61.7** | **58.0** | **62.3** | **52.5** | **51.0** | **39.7** | 57.2 | **60.6** | **51.4** | 32.8 | **55.0** | **38.6** | **27.6** | 35.6 | **48.9** |
| RoTTA | ✗ | 68.0 | 62.4 | 68.6 | 58.5 | 56.7 | 40.9 | **56.6** | 60.7 | 52.2 | 30.2 | 60.7 | 39.7 | 27.8 | **35.1** | 51.3 |
| | ✓ | 69.0 | 62.8 | 68.9 | 58.1 | 58.0 | 40.9 | 57.2 | 61.0 | 53.0 | **30.0** | 62.4 | 41.2 | 27.7 | 35.8 | 51.9 |

Table 3: **Effect of Adversarial Memory Initialization on TTA in Tracklet-Wise *non*-i.i.d. Scenario.** On ITD, RoTTA with ADVMEM (✓) significantly outperforms standard memory (✗), highlighting its effectiveness in non-i.i.d. tracklet settings.

| Method | ADVMEM | Noise | | | Blur | | | Weather | | | | Digital | | | | Avg. |
|---|---|---|---|---|---|---|---|---|---|---|---|---|---|---|---|---|
| | | gauss. | shot | impul. | defoc. | glass | zoom | snow | frost | fog | brigh. | contr. | elast. | pixel. | jpeg | |
| TENT | ✗ | 94.2 | 94.3 | 94.3 | 94.5 | 94.5 | 94.0 | 93.7 | 93.6 | 93.4 | 93.0 | 92.9 | 93.8 | 93.4 | 93.6 | 93.8 |
| | ✓ | 89.2 | 89.8 | 92.4 | 92.1 | 92.4 | 86.6 | 91.4 | 90.2 | 90.7 | 83.7 | 83.9 | 89.0 | 79.5 | 81.5 | 88.0 |
| SHOT-IM | ✗ | 93.9 | 93.8 | 93.9 | 94.3 | 94.3 | 93.7 | 93.4 | 93.2 | 92.7 | 92.5 | 92.5 | 93.6 | 93.5 | 93.2 | 93.6 |
| | ✓ | 92.3 | 92.2 | 92.7 | 92.4 | 93.1 | 91.6 | 92.5 | 91.8 | 91.5 | 90.5 | 92.0 | 91.6 | 89.1 | 90.9 | 91.7 |
| RoTTA | ✗ | 85.6 | **81.6** | 86.5 | 86.9 | 87.3 | 78.9 | 81.7 | 83.5 | 75.0 | 67.2 | **71.8** | 79.7 | 77.1 | 67.4 | 79.3 |
| | ✓ | **84.4** | 82.2 | **84.1** | **83.0** | **83.3** | **68.6** | **80.2** | **80.5** | **74.7** | **62.8** | 73.6 | **75.2** | **61.3** | **62.9** | **75.5** |

where $y$ is a randomly assigned label, and $\mathcal{L}_{ce}$ is the cross-entropy loss. We solve this optimization problem by applying gradient descent, starting from Gaussian noise. This process is repeated $N$ times to fully initialize $\mathcal{M}$. We term this procedure "ADVMEM" and present it in Algorithm 1.

ADVMEM provides two main benefits: **(i)** it keeps the memory bank populated with class-representative samples, reducing forgetting under strong non-i.i.d. streams, and **(ii)** it is independent of how $\mathcal{M}$ is updated or used—e.g., applying it to RoTTA does not alter the adaptation mechanism. A simpler alternative, initializing $\mathcal{M}$ with uniformly sampled, non-corrupted training examples, was tested but (see appendix) did not improve performance even when privacy is not a concern.

---

**Algorithm 1** ADVMEM

**function** INITIALIZEMEMORY($f_\theta, K, N$)
    Initialize $\mathcal{M} = \{\}$
    **while** $|\mathcal{M}| < N$ **do**
        $x \sim \mathcal{N}(0, I), y \sim \mathcal{U}\{1, 2, \ldots, K\}$
        **while** $f_\theta(x) \neq y$ **do**
            $x \leftarrow x - \alpha \cdot (\nabla_x \mathcal{L}_{ce}(f_\theta(x), y))$
        **end while**
        $\mathcal{M} \leftarrow \mathcal{M} \cup x$
    **end while**
    **return** $\mathcal{M}$
**end function**

---

**Tracklet-wise i.i.d. Setup:** We first analyze the Tracklet-wise i.i.d. setting from Section 4.6 (i.e., $\gamma \to \infty$). Table 2 reports the results, showing significant performance improvements across all baselines. Notably, ADVMEM reduces the average error rate of TENT by ∼15% and that of SHOT-IM by over 40%, making SHOT-IM the top-performing method. These improvements highlight the effectiveness of ADVMEM in stabilizing adaptation.

**Tracklet-wise non-i.i.d. Setup:** To further examine its impact, we evaluate ADVMEM in the extreme non-i.i.d. setting from Section 4.7 ($\gamma \to 0$). Table 3 presents the results, demonstrating substantial performance gains. In particular, ADVMEM enhances RoTTA's performance by an average of 4%, with specific improvements of 16% and 10% against pixelate and zoom corruptions, respectively. This trend holds for other methods as well, with TENT improving by over 10% on JPEG corruption and achieving a 5% average improvement across all corruptions.

# 6 ABLATION STUDIES AND ANALYSIS

This section presents an in-depth analysis of the ITD dataset and our proposed ADVMEM. We evaluate ResNet-18 and Vision Transformer (ViT).

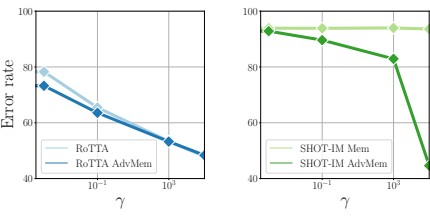 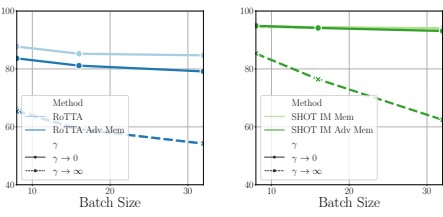

(a) Transitioning from *non*-i.i.d. to i.i.d..    (b) Influence of batch size.

Figure 4: **Error rate as (a)** evaluation shifts from non-i.i.d. to i.i.d., and **(b)** batch size varies. In **(a)**, we control label distribution i.i.d.-ness via the $\gamma$ parameter in $\text{Dir}(\gamma)$ (log-scale). ADVMEM consistently improves or maintains performance across regimes. In **(b)**, larger batch sizes improve performance, further boosted by ADVMEM. All results use ResNet-18.

## 6.1 Controlling Label Distribution

Using a Dirichlet distribution $\text{Dir}(\gamma)$ for sampling labels, as introduced in Yuan et al. (2023), allows control over the distributional shift in the label space through the parameter $\gamma$. Specifically, as $\gamma \to 0$, we approach a class-incremental non-i.i.d. setup, while as $\gamma \to \infty$, we transition towards a uniform i.i.d. setup. We extend our experiments by analyzing intermediate stages with $\gamma \in \{10^{-4}, 10^{-1}, 10^3\}$ and report the results in Figure 4a.

For low values of $\gamma$, the model encounters a challenging class-incremental non-i.i.d. scenario. In this case, ADVMEM proves instrumental in mitigating class bias within the incoming image stream, reducing degradation in performance. Conversely, as $\gamma$ increases, the scenario becomes easier, as the model has higher chances of encountering uniformly sampled classes. In this context, the adversarially initialized memory samples are rapidly replaced by reliable examples from the stream, making the initialization inconsequential. Consequently, ADVMEM neither enhances nor degrades performance in the uniform i.i.d. setup.

## 6.2 Sensitivity to Batch Size

We analyze the impact of batch size on performance by varying it across $\{8, 16, 32\}$. The results reported in Figure 4b confirm that increasing the batch size improves performance, as larger batches expose models to more diverse examples, facilitating better adaptation.

When comparing i.i.d. and non-i.i.d. setups ($\gamma \to \infty$ vs. $\gamma \to 0$), we observe that methods incorporating ADVMEM consistently achieve improved or maintained performance across batch sizes. However, as in Section A.3, the impact of ADVMEM is less pronounced in the i.i.d. setup, indicating that memory initialization has a limited effect in this scenario. These insights highlight the robustness of ADVMEM in highly non-i.i.d. environments while demonstrating that its advantages diminish in simpler, uniform settings. Overall, the findings reinforce the necessity of well-designed memory initialization strategies in real-world, dynamically shifting data distributions.

## 7 Conclusion

This work introduces ITD, a benchmark capturing the temporal dependencies of real-world data streams, challenging existing TTA methods often evaluated on simplified settings. We also propose ADVMEM, an adversarial memory initialization strategy that mitigates forgetfulness and enhances adaptability, particularly in non-i.i.d. scenarios. Our results highlight the need for benchmarks and adaptation strategies that reflect evolving data distributions, guiding future research toward more resilient and realistic TTA methods.

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

## A  ITD Construction Analysis & Experimental Setup Details

### A.1  Dataset Distribution Overview

The dataset distribution of the 21 object classes, as shown in Figure 5, presents a varied representation in terms of the number of objects and instances across different classes. The distribution is not uniform, with some classes having a higher number of instances compared to others. This variation provides a diverse range of object occurrences, which can be reflective of different scenarios where certain objects appear more frequently while others are less common. Such a distribution can be valuable in assessing model performance across a broad spectrum of object categories, ensuring that both commonly and less commonly occurring objects are adequately represented in the dataset.

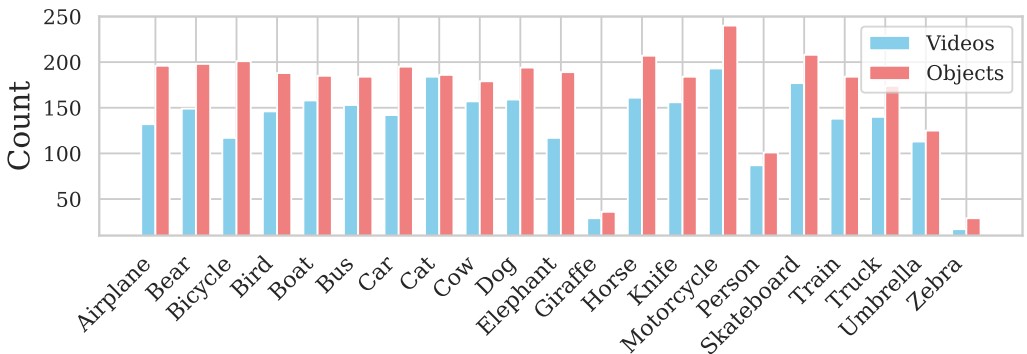

Figure 5: **Class distribution of the test set.** Here we report a detailed breakdown of the distribution of the 21 object classes in terms of the number of objects and instances.

### A.2  Experiment Setup Overview

Figure 3 provides a visual representation of the two distinct experimental setups used in our study: the frame-wise and tracklet-wise configurations. In the frame-wise setup, a single frame is sampled from each tracklet, ensuring that each object is observed only once. This setup allows for a broad, non-redundant sampling of the dataset, suitable for scenarios where each object instance is considered independently. On the other hand, the tracklet-wise setup processes the frames within each tracklet sequentially, capturing the temporal continuity and variations that occur as the object is observed over time. Both setups are further divided into i.i.d. (independent and identically distributed) and non-i.i.d. settings, providing a comprehensive evaluation framework. The i.i.d. setting assumes that the frames or tracklets are sampled without any dependency, simulating a random observation scenario. Conversely, the non-i.i.d. setting introduces dependencies between samples, reflecting more realistic conditions where observations are not entirely independent, such as in continuous video streams. This dual approach allows for a thorough assessment of model performance under varying assumptions of data distribution and sampling.

### A.3  Controlling the Label Distribution in Streaming Scenarios

Using a Dirichlet distribution $\text{Dir}(\gamma)$ for sampling labels, as introduced in Yuan et al. (2023), provides a mechanism for controlling the distribution shift in the label space through the distribution's $\gamma$ parameter.

Specifically, as $\gamma \to 0$, we converge to a class-incremental non-i.i.d. setup, while, as $\gamma \to \infty$, we transition towards a uniform i.i.d. setup. We thus extend our experiments by analyzing intermediate stages with $\gamma \in \{10^{-4}, 10^{-1}, 10^3\}$, and report the results in Figure 4a.

For low values of $\gamma$, the model observes a challenging class-incremental non-i.i.d. scenario. In this evolving context, our proposed memory initialization technique proves instrumental in enhancing

Figure 6: **Dynamic Severity:** We consider temporal variations in the severity of corruptions, reflecting the realistic scenarios where the impact of corruptions may change over time within a single video clip.

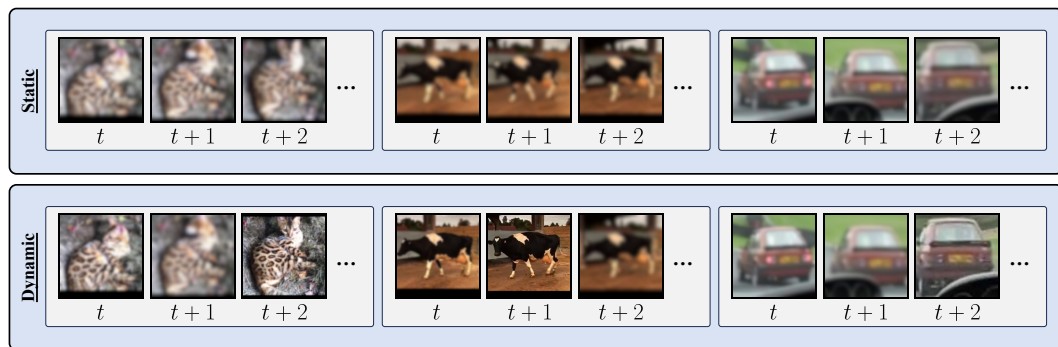

model performance. The adversarially initialized memory addresses class bias within the incoming image stream and mitigates degradation in performance.

Conversely, as $\gamma$ increases, the scenario becomes easier, as the model has higher chances of encountering images from uniformly-sampled classes. In this scenario, the memory samples that were initialized by ADVMEM are likely to be quickly and completely replaced with reliable samples from the stream. This quick replacement is a result of the stream's extreme uniformity, which causes any initialization to be inconsequential. In this context, ADVMEM neither augments nor diminishes performance.

### A.4 SENSITIVITY TO BATCH SIZE IN STREAMING ADAPTATION

In this section, we explore the impact of batch size on performance. In particular, we vary the batch size in $\{8, 16, 32\}$, and report our results in Figure 4b. As expected, increasing the batch size improves performance, since larger batches provide models with more (and more diverse) examples on which to adapt. When comparing between the i.i.d. and non-i.i.d. setups (*i.e.* $\gamma \to \infty$ *vs.* $\gamma \to 0$), we find that methods equipped with ADVMEM exhibit improved or at least sustained performance across batch sizes. However, similar to our observations in previous sections, the impact of ADVMEM is less pronounced in the i.i.d. setting ($\gamma \to \infty$), indicating that memory initialization has limited effect in such scenarios.

As mentioned in previous section, we use a default batch size of 64. This batch size ensures that exactly one tracket (64 frames) fits in one forward pass. In previous sections, we experiment with different batch sizes, which are smaller than the default. When the batch size is smaller than the number of frames in a tracklet, we adopt a sequential processing approach, where a tracklet is consecutively processed until all frames are observed. This sequential approach ensures that the entire content of a tracklet is leveraged, potentially enabling the model to adapt effectively to the inherent temporal dependencies and patterns within the data.

However, counterintuitively, we can see from Fig. 4 that smaller batch sizes do not lead to a lower error rate in our experiments. While sequential processing allows for a detailed examination of temporal intricacies within a tracklet, the models, influenced by label distribution, exhibit a degradation in performance with smaller batch sizes. This observation underscores the intricate interplay between batch size, sequential information utilization, and the model's robustness to label distribution.

Importantly, our proposed memory initialization (ADVMEM) consistently improves or maintains performance compared to standard memory initialization, regardless of batch size. This fact highlights the robustness and effectiveness of ADVMEM in diverse experimental conditions.

Table 4: **Effect of Adversarial Memory Initialization on TTA performance under Dynamic Severity:** We assess the impact of memory initialization techniques on TTA methods in the dynamic severity setting, where the intensity of corruptions varies within the tracklet. The table presents results for standard (✗) and adversarial (✓) memory initialization (ADVMEM).

(a) Tracklet wise i.i.d.

| Method | ADVMEM | Noise | | | Weather | | | Avg. |
|--------|--------|-------|------|-------|-------|------|--------|------|
| | | gauss. | shot | impul. | frost | fog | brigh. | |
| TENT | ✗ | 89.4 | 89.6 | 91.3 | 92.3 | 91.8 | 92.3 | 91.1 |
| | ✓ | 68.5 | 69.0 | 70.6 | 81.2 | 73.4 | 56.1 | 69.8 |
| SHOT-IM | ✗ | 89.7 | 89.2 | 91.2 | 92.1 | 91.9 | 92.2 | 91.0 |
| | ✓ | **38.4** | **38.1** | **45.5** | 53.3 | 40.3 | 25.0 | 40.1 |
| RoTTA | ✗ | 41.1 | 38.1 | 50.4 | **49.4** | **38.3** | **23.0** | **40.0** |
| | ✓ | 40.7 | 38.3 | 50.0 | 49.6 | 39.1 | 23.4 | 40.2 |

(b) Tracklet wise *non*-i.i.d.

| Method | ADVMEM | Noise | | | Weather | | | Avg. |
|--------|--------|-------|------|-------|-------|------|--------|------|
| | | gauss. | shot | impul. | frost | fog | brigh. | |
| TENT | ✗ | 89.5 | 89.8 | 91.3 | 92.3 | 91.9 | 92.3 | 91.2 |
| | ✓ | 78.1 | 78.1 | 82.7 | 87.3 | 84.7 | 78.8 | 81.6 |
| SHOT-IM | ✗ | 89.8 | 88.9 | 91.2 | 92.3 | 91.9 | 92.1 | 91.0 |
| | ✓ | 89.8 | 89.7 | 91.4 | 91.1 | 90.9 | 89.8 | 90.4 |
| RoTTA | ✗ | 61.2 | 62.1 | 68.8 | 75.1 | 66.9 | 59.4 | 65.6 |
| | ✓ | **58.6** | **60.3** | **67.2** | **74.7** | **66.1** | **51.1** | **63.0** |

## B DYNAMIC CORRUPTION INCORPORATION: ANALYSIS AND RESULTS

Incorporating dynamic corruptions, as illustrated in Figure 6, into our experiments involves the continuous application of corruptions within the tracklet, where the severity level is defined as a function of time. This dynamic approach enables us to precisely control the severity level of each corruption, closely mimicking real-world scenarios. For example, when simulating defocus blur, the dynamic corruption setting introduces fluctuations in focus, alternating between in and out of focus. Similarly, for weather-related corruptions, such as rain, the dynamic application varies the intensity over time, simulating realistic variations in weather conditions within the video clips. Dynamic corruptions are controlled by the severity function $S(t) = s \cdot |\text{sign}(t)|$, where $s$ represents the severity level and $t$ is the index of the frame in a given tracklet. In contrast to static setups, each frame $k_t$ of the $k$-th tracklet experiences variable severity $S(t)$ at time $t$. The severity function can be customized for each corruption type by adjusting the function's parameters (*i.e.* frequency) or additional factors such as random noise. As depicted in Table 4, the deployment of our proposed ADVMEM in the context of dynamic corruptions exhibits a consistent trend, akin to our findings from the experiments on static corruptions. This observation underscores that the utilization of ADVMEM consistently again enhances or maintains performance across diverse scenarios.

## C ADDITIONAL ABLATIONS: VISION TRANSFORMER (VIT) EXPERIMENTS

We extend our experiments to include the Vision Transformer (ViT) architecture. ViT outperforms ResNet-18, even at lower batch sizes, due to its reduced sensitivity to batch size Niu14 et al.. Our ViT experiments focus on the impact of varying batch sizes on method performance (Figure 7). Larger batch sizes do consistently boost performance, with ADVMEM adding further improvements. This analysis highlights the adaptability and effectiveness of ADVMEM.

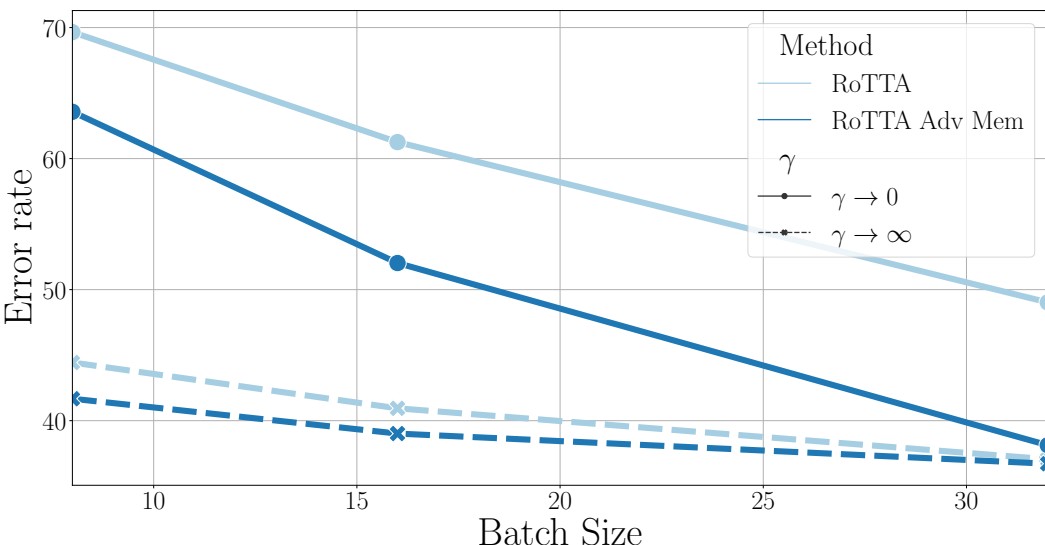

Figure 7: **Error rate as we vary the batch size.** We study the influence of batch size on method performance. Larger batch sizes enhance performance across the board, with our proposed ADVMEM consistently contributing to further improvements. All results presented here are for ViT-B-16.

## D    EVALUATION OF MEMORY INITIALIZATION USING TRAINING SAMPLES

---
**Algorithm 2** TrainMem

---
    **function** INITIALIZEMEMORY($K, N$)
        Initialize $\mathcal{M} = \{\}$
        **while** $|\mathcal{M}| < N$ **do**
            $y \sim \mathcal{U}\{1, 2, \ldots, K\}$
            $x \sim \mathcal{U}(\mathcal{D}\{x|y\})$
            $\mathcal{M} \leftarrow \mathcal{M} \cup x$
            $\mathcal{D} \leftarrow \mathcal{D} \setminus \{x\}$
        **end while**
        **return** $\mathcal{M}$
    **end function**

---

In contrast to the approach outlined in Algorithm 1, where adversarial samples are employed, Algorithm 2 initializes the memory with images from the training set, denoted by $\mathcal{D}$. Here, $\mathcal{D}$ consists of images and their corresponding labels. For any label $y$, $\mathcal{D}\{x|y\}$ represents the subset of $\mathcal{D}$ corresponding to images $x$ labeled as $y$. The initialization process involves uniformly selecting images from $\mathcal{D}$ without replacement until the memory is full. This procedure, which we refer to as Train-Mem, results in a class-wise balanced memory initialization, similar to our adversarial one.

Table 5 summarizes the results for the tracklet-wise non-i.i.d. setup. We observe that while initializing the memory with TrainMem, *i.e.* via Algorithm 2, has positive impact in reducing the error rate of RoTTA, it significantly underperforms our novel ADVMEM. For example, TrainMem reduces the error rate against glass blur by 1.7% when compared to RoTTA. However, ADVMEM improves over this naive initialization by over 2% against the same corruption.

## E    VISUALIZING ADVERSARIAL EXAMPLES FOR ADVMEM INITIALIZATION

In this section, we present visualizations of adversarial examples used for initializing ADVMEM. These examples are generated during the memory bank initialization process. For details on the

Table 5: **Tracklet-wise *non*-i.i.d.:** Assessment of TTA methods on ITD in a *non*-i.i.d. tracklet context. We contrast standard memory initialization (✗), memory initialized with training samples (✗), and adversarial memory initialization (✓).

| Method | ADVMEM | Noise | | | Blur | | | Weather | | | | Digital | | | | Avg. |
|--------|--------|-------|------|--------|-------|-------|------|------|-------|------|--------|-------|--------|--------|------|------|
| | | gauss. | shot | impul. | defoc. | glass | zoom | snow | frost | fog | brigh. | contr. | elast. | pixel. | jpeg | |
| | ✗ | 85.6 | **81.6** | 86.5 | 86.9 | 87.3 | 78.9 | 81.7 | 83.5 | 75.0 | 67.2 | 71.8 | 79.7 | 77.1 | 67.4 | 79.3 |
| RoTTA | ✗ | **83.0** | **81.6** | **83.1** | 86.3 | 85.6 | 77.3 | 84.4 | **79.7** | 75.4 | 71.0 | **69.5** | 78.3 | 71.8 | 67.7 | 78.2 |
| | ✓ | 84.4 | 82.2 | 84.1 | **83.0** | **83.3** | **68.6** | **80.2** | 80.5 | **74.7** | **62.8** | 73.6 | **75.2** | **61.3** | **62.9** | **75.5** |

Table 6: **Performance Comparison under Frame-wise i.i.d. Assumption.** We report average error rates for TTA methods on images from our ITD dataset. In this setup, all images/frames are shuffled and then independently subjected to corruptions. Notably, SHOT-IM outperforms all other methods and across all corruptions, showcasing its robustness against these domain shifts.

| Method | Noise | | | Blur | | | | Weather | | | | Digital | | | | Avg. |
|--------|-------|------|--------|-------|-------|--------|------|------|-------|------|--------|-------|--------|--------|------|------|
| | gauss. | shot | impul. | defoc. | glass | motion | zoom | snow | frost | fog | brigh. | contr. | elast. | pixel. | jpeg | |
| Source | 93.8 | 90.8 | 94.0 | 50.8 | 45.1 | 51.2 | 45.2 | 61.7 | 70.4 | 70.6 | 31.4 | 86.6 | 64.0 | 39.1 | 41.1 | 62.4 |
| AdaBN | 57.9 | 56.0 | 58.2 | 50.8 | 51.7 | 44.2 | 37.9 | 53.0 | 55.7 | 50.6 | 32.5 | 50.8 | 37.8 | 29.0 | 36.8 | 46.9 |
| SHOT-IM | **47.5** | **45.5** | **47.1** | **43.6** | **43.2** | **36.4** | **31.2** | **47.1** | **50.2** | **41.7** | **27.8** | **44.5** | **29.6** | **24.6** | **30.0** | **39.3** |
| TENT | 57.9 | 55.8 | 58.4 | 50.6 | 51.7 | 44.2 | 37.8 | 52.9 | 55.6 | 50.4 | 32.5 | 50.4 | 37.7 | 28.9 | 36.8 | 46.8 |
| SAR | 57.6 | 55.3 | 59.1 | 50.4 | 51.3 | 43.7 | 37.6 | 52.8 | 55.4 | 50.1 | 32.2 | 49.4 | 37.6 | 28.6 | 36.3 | 46.5 |
| CoTTA | 57.9 | 55.8 | 58.6 | 50.3 | 51.3 | 43.8 | 37.6 | 53.0 | 55.8 | 50.4 | 32.3 | 50.4 | 37.8 | 28.9 | 36.7 | 46.7 |
| ETA | 57.9 | 54.6 | 59.0 | 50.5 | 51.6 | 44.0 | 37.5 | 53.1 | 55.7 | 50.7 | 32.4 | 50.9 | 37.4 | 28.8 | 36.7 | 46.7 |
| EATA | 57.9 | 55.1 | 58.1 | 50.2 | 51.9 | 43.4 | 38.0 | 54.4 | 56.2 | 50.9 | 31.9 | 50.3 | 38.8 | 28.5 | 36.7 | 46.8 |
| RoTTA | 69.4 | 64.0 | 71.2 | 54.4 | 55.0 | 50.6 | 42.4 | 60.0 | 62.0 | 59.6 | 34.3 | 64.3 | 43.3 | 31.1 | 38.7 | 53.4 |

Table 7: **Tracklet-wise i.i.d. Evaluation with and without Memory.** We report average error rates for TTA methods on our ITD dataset under the tracklet-wise i.i.d. scenario (*i.e.* entire tracklets are sampled i.i.d.). The results are grouped to reflect the presence (✓) or absence (✗) of memory during adaptation. RoTTA shows notable robustness by using memory, significantly outperforming the non-memory variants across various corruption types, as demonstrated by the marked difference in error rate.

| Method | Memory | Noise | | | Blur | | | Weather | | | | Digital | | | | Avg. |
|--------|--------|-------|------|--------|-------|-------|------|------|-------|------|--------|-------|--------|--------|------|------|
| | | gauss. | shot | impul. | defoc. | glass | zoom | snow | frost | fog | brigh. | contr. | elast. | pixel. | jpeg | |
| TENT | ✗ | 94.3 | 94.3 | 94.4 | 94.5 | 94.6 | 94.1 | 94.1 | 94.5 | 93.4 | 93.1 | 93.7 | 94.0 | 93.5 | 93.7 | 94.0 |
| | ✓ | 94.2 | 94.2 | 94.3 | 94.5 | 94.5 | 93.9 | 93.7 | 93.7 | 93.2 | 93.0 | 92.8 | 93.8 | 93.4 | 93.5 | 93.8 |
| SHOT-IM | ✗ | 94.8 | 94.7 | 94.8 | 94.8 | 94.7 | 94.8 | 94.7 | 94.6 | 94.6 | 94.4 | 94.7 | 94.9 | 94.7 | 94.8 | 94.7 |
| | ✓ | 93.8 | 93.9 | 93.9 | 94.2 | 94.3 | 93.7 | 93.3 | 93.0 | 93.0 | 92.8 | 92.1 | 93.5 | 93.4 | 93.2 | 93.4 |
| RoTTA | ✓ | **68.0** | **62.4** | **68.6** | **58.5** | **56.7** | **40.9** | **56.6** | **60.7** | **52.2** | **30.2** | **60.7** | **39.7** | **27.8** | **35.1** | **51.3** |

Table 8: **Tracklet-wise *non*-i.i.d. Evaluation with and without Memory Banks.** We report the performance of TTA methods when tracklets are non-i.i.d., *i.e.* tracklets are sampled such that their labels display correlation in time (by following a Dirichlet distribution). We further examine whether using a memory bank (✓) influences outcomes. RoTTA with memory exhibits the lowest error rates, indicating proficiency against non-i.i.d. data. Without memory (✗), all methods experience worse error rates, underscoring the impact of memory in adapting to complex data streams.

| Method | Memory | Noise | | | Blur | | | Weather | | | | Digital | | | | Avg. |
|--------|--------|-------|------|--------|-------|-------|------|------|-------|------|--------|-------|--------|--------|------|------|
| | | gauss. | shot | impul. | defoc. | glass | zoom | snow | frost | fog | brigh. | contr. | elast. | pixel. | jpeg | |
| TENT | ✗ | 94.3 | 94.3 | 94.4 | 94.5 | 94.6 | 94.0 | 94.0 | 95.0 | 93.5 | 93.2 | 93.5 | 94.0 | 93.5 | 93.7 | 94.0 |
| | ✓ | 94.2 | 94.3 | 94.3 | 94.5 | 94.5 | 94.0 | 93.7 | 93.6 | 93.4 | 93.0 | 92.9 | 93.8 | 93.4 | 93.6 | 93.8 |
| SHOT-IM | ✗ | 95.1 | 94.9 | 95.1 | 95.1 | 95.1 | 95.7 | 94.9 | 95.2 | 94.8 | 95.0 | 95.0 | 94.8 | 95.3 | 95.2 | 95.1 |
| | ✓ | 93.9 | 93.8 | 93.9 | 94.3 | 94.3 | 93.7 | 93.4 | 93.6 | 93.2 | 92.7 | 92.5 | 93.6 | 93.5 | 93.2 | 93.6 |
| RoTTA | ✓ | **85.6** | **81.6** | **86.5** | **86.9** | **87.3** | **78.9** | **81.7** | **83.5** | **75.0** | **67.2** | **71.8** | **79.7** | **77.1** | **67.4** | **79.3** |

creation of adversarial examples, please refer to ADVMEM sections. Figure 8 showcases a selection of these adversarial examples.

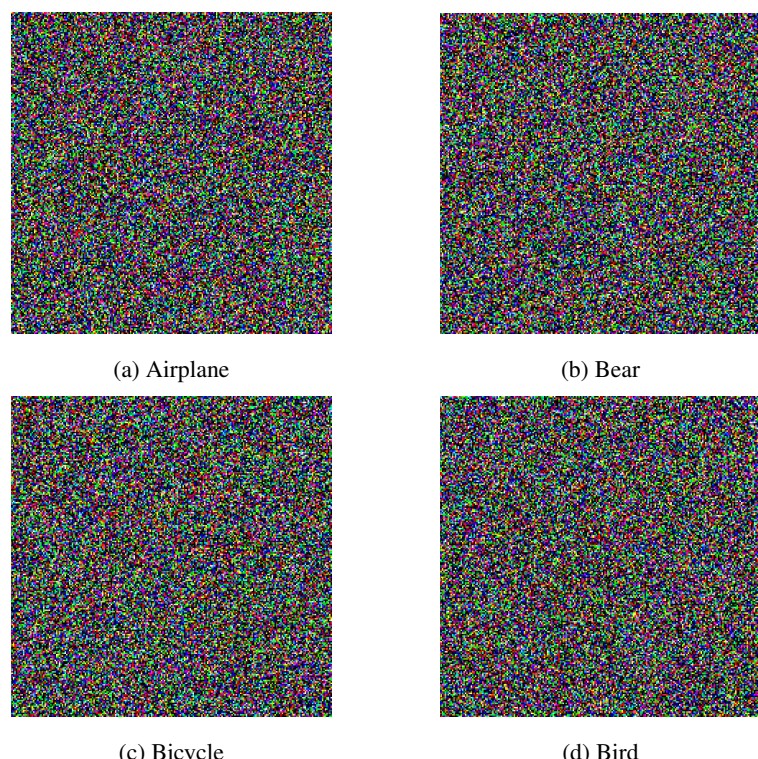

(a) Airplane

(b) Bear

(c) Bicycle

(d) Bird

Figure 8: Visualizations of selected adversarial examples used for initializing the memory bank in ADVMEM.

