# OpenReview forum: "ADVMEM: Adversarial Memory Initialization for Realistic Test-Time Adaptation via Tracklet-Based Benchmarking"
_ICLR.cc/2026/Conference — Submitted to ICLR 2026_

### Official Review · Reviewer_urTJ · 2025-10-28

**Soundness:** 3
**Presentation:** 3
**Contribution:** 1
**Rating:** 4
**Confidence:** 5

**Summary:**

This paper introduces a new dataset, to improve Test-Time Adaptation (TTA)—where models adapt to distribution shifts during inference—under realistic temporal dependencies found in video streams. It introduces both synthetic and realistic evaluation settings upon a medium-sized dataset comprised of 34k objects and 21 objects.

**Strengths:**

Realistic Benchmark Construction:
* The curated dataset seems reasonable and solid.

**Weaknesses:**

Additional baselines
* The authors should consider https://openreview.net/forum?id=E9HNxrCFZPV - a TTA method tackling extremely relevant scenario & scheme.
Questions upon novelty.
* There are mutliple works tackling, specifically upon temporal correlation settings within TTA. To name a few, NOTE, https://openreview.net/forum?id=xyxU99Nutg, etc.
* However, the authors did not include nor compare their settings with the scenarios proposed in these papers.
* To me, this benchmark seems more like of a extension, a realization of the proposed synthetic tasks. Thus, there are multiple ongoing works already trying to solve the issue, and this is yet another benchmark that re-identifies a pre-existing problem. Can the authors provide a sufficient argument on how their method raises NEW problems that current TTA methods fail? If this is resolved, I am willing to raise my score.

**Questions:**

None.

---

### Official Review · Reviewer_Hfyn · 2025-10-29

**Soundness:** 3
**Presentation:** 2
**Contribution:** 2
**Rating:** 2
**Confidence:** 4

**Summary:**

The paper introduces a tracklet-based benchmark, ITD, built from TrackingNet to study TTA under temporal correlations. Moreover, the authors propose an ADVMEM method to initialize the memory bank and show improvement over a memory-based TTA method. However, I raise point-by-point concerns regarding the manuscript’s contributions below, and thus recommend Reject.

**Strengths:**

1.	The paper introduces the ITD dataset from realistic videos with inherent temporal correlation, and explores both static and dynamic corruption severities within the tracklet.
2.	The proposed ADVMEM method is simple to follow.

**Weaknesses:**

1.    Unconvincing novelty against existing benchmarks. In TTA for semantic segmentation and object detection, established benchmarks like ACDC, CityScape-C, and KITTI-C also exhibit temporal dependencies and distribution shifts, whereas the introduced ITD merely appears to be a simplification of these benchmarks with a single object in each frame.
2.	Limited evaluation of TTA approaches. The most recent TTA work discussed or compared in the manuscript is from 2023. To establish a comprehensive evaluation as claimed, it is suggested to analyse with state-of-the-art TTA methods, such as ROID [1], CMF [2], DeYO [3], BeCoTTA [4],  POEM [5], PeTTA [6], DPCore [7], ReCAP [8], TTE [9], and EATA-C [10]  to study how existing TTA methods perform on the introduced ITD dataset.
3.	Lacking appropriate and diverse base models for evaluating TTA methods on the ITD dataset. Despite introducing the ITD dataset, there are no suitable base models for rigorous evaluation on the dataset. The authors finetune a ResNet18 and a ViT-B model with a limited budget on the ITD dataset. However, as evidenced by Table 4, prior TTA methods such as EATA/SAR/CoTTA achieve no better results than AdaBN with the finetuned ResNet18, suggesting that the limitedly finetuned model may not faithfully assess different TTA methods. Moreover, a larger coverage regarding model sizes (e.g., ViT-S/B/L) and architectures (e.g., ConvNext, CLIP) is recommended.
4.	Lack of explanations/insights into how ADVMEM outperforms the warmup strategy with training data. While the authors provide some empirical results, no further explanations or analysis are given. It is thus infeasible to judge whether such enhancements are specific to training strategies and datasets.
5.	Writing can be improved. This mainly includes two aspects: 1) Formatting: the organization of the introduction, the format of citation, and also the format of some tables (e.g., Tables 4 & 5) seems non-standard or inappropriate. 2) Clarity, such as detailing hyperparameters used for each TTA method, introducing a self-contained proof for the hypothesis of the correlation between empty initialization and catastrophic forgetting.

I believe that the manuscript can be significantly improved by offering diverse models finetuned/trained for the ITD dataset, broadening the evaluation with more state-of-the-art TTA methods, and digging into why ADVMEM is an elegant and essential technique for robust TTA in the wild. The current manuscript falls short of the standard for ICLR, and I thus recommend Reject, but believe that incorporating the modifications as suggested above could make the manuscript a strong contribution to the field, and wish the authors the best.

[1] Universal test-time adaptation through weight ensembling, diversity weighting, and prior correction. WACV 2024.

[2] Continual momentum filtering on parameter space for online test-time adaptation. ICLR 2024.

[3] Entropy is not enough for test-time adaptation: From the perspective of disentangled factors. ICLR 2024.

[4] BECoTTA: Input-dependent Online Blending of Experts for Continual Test-time Adaptation. ICML 2024.

[5] Protected test-time adaptation via online entropy matching: A betting approach. NeurIPS 2024.

[6] Persistent Test-Time Adaptation in Recurring Testing Scenarios. NeurIPS 2024.

[7] Dpcore: Dynamic Prompt Coreset for Continual Test-Time Adaptation. ICML 2025.

[8] Beyond entropy: Region confidence proxy for wild test-time adaptation. ICML 2025.

[9] Test-time ensemble via linear mode connectivity: A path to better adaptation. ICLR 2025.

[10] Uncertainty-Calibrated Test-Time Model Adaptation without Forgetting. TPAMI 2025.

**Questions:**

See weaknesses above.

---

### Official Review · Reviewer_AaSt · 2025-10-30

**Soundness:** 2
**Presentation:** 2
**Contribution:** 2
**Rating:** 4
**Confidence:** 4

**Summary:**

This paper introduces ITD, a benchmark for online test-time adaptation based on temporally correlated tracklets with controlled class skew and synthetic corruptions. It enables fine-grained evaluation of TTA methods under more realistic streaming conditions and reveals limitations in existing adaptation approaches.

**Strengths:**

1. The paper introduces a novel benchmark for test-time adaptation using temporally correlated image tracklets with controlled class skew and clean-to-corrupt transitions, enabling evaluation of model robustness under realistic online drift conditions.

2. The dataset design isolates specific shift factors in a modular way, allowing for fine-grained ablation and analysis of TTA methods beyond what existing benchmarks like ImageNet-C or CIFAR 10-C/CIFAR 100-C can support.

**Weaknesses:**

1. The dataset crops and resizes bounding boxes to a fixed size, removing natural scale variation. This leads to unrealistic object representations, especially problematic in applications where object size carries semantic meaning, like autonomous driving.

2. Tracklets contain only minor appearance changes and lack meaningful motion or occlusion, limiting the dataset’s ability to evaluate temporal adaptation methods that require modeling strong time dynamics.

3. Only 21 object classes are included, restricting the benchmark’s coverage and limiting its utility in long-tailed or open-world scenarios where rare classes are common.

4. The corruptions are synthetic and do not enough to reflect real-world sensor degradations, reducing ecological validity.

5. The dataset is vision-only. Many recent TTA applications, especially with VLMs like CLIP or BLIP, involve multimodal inputs. The lack of modality diversity limits relevance to current research.

6. Since 2024, many temporal and memory-based TTA methods have emerged, especially for continual and streaming settings [1]. The benchmark does not reflect these developments.

[1] In Search of Lost Online Test-time Adaptation: A Survey

**Questions:**

See weaknesses part.

---

### Official Review · Reviewer_abTe · 2025-11-04

**Soundness:** 2
**Presentation:** 2
**Contribution:** 2
**Rating:** 2
**Confidence:** 4

**Summary:**

This paper introduces ITD (Inherent Temporal Dependencies), a new benchmark for evaluating Test-Time Adaptation (TTA) under realistic temporal correlations by constructing tracklet-based datasets from TrackingNet. It argues that existing TTA benchmarks fail to model temporal dependencies and proposes ADVMEM, an adversarial memory initialization strategy to improve memory-based TTA performance. Experiments on ITD suggest that ADVMEM stabilizes adaptation and reduces error rates under both i.i.d. and non-i.i.d. scenarios.

**Strengths:**

1. The paper correctly identifies that most existing TTA evaluations assume i.i.d. data and ignore temporal correlations, a real-world challenge in streaming and video-based settings.
2. Using object tracklets from TrackingNet to build ITD is a reasonable and practical idea to simulate non-i.i.d. data streams.

**Weaknesses:**

1. The proposed ADVMEM is a rather straightforward application of adversarial sample generation for memory initialization, not a fundamentally new TTA mechanism. The connection between adversarial initialization and improved adaptation is not theoretically analyzed, only empirically observed.
2. ITD largely reuses existing datasets (TrackingNet + corruptions), without strong justification for its novelty compared to prior benchmarks like RoTTA or ImageNet-Vid. The paper does not release or sufficiently describe the dataset splits, reproducibility details, or ethical considerations.
3. The paper is long, repetitive, and poorly formatted, with significant redundancy across sections and appendices. Figures and tables are not well integrated into the narrative, captions often repeat content. The title and abstract oversell the “adversarial” and “realistic” aspects without corresponding depth.
4. There’s little analysis explaining why ADVMEM helps or how the adversarial initialization affects feature representations. The benchmark’s diversity and scalability are not convincingly analyzed.

**Questions:**

1. How does ITD differ from existing video-based TTA or non-i.i.d. continual adaptation datasets (e.g., ImageNet-Vid, UCF101 streams, RoTTA and TRIBE setups)?

---

### Meta-Review · Area_Chair_ELiE · 2026-01-11

**Summary:**

This submission proposes ITD, a tracklet-based benchmark for test-time adaptation under temporal dependencies, and ADVMEM, an adversarial memory initialization intended to improve memory-based TTA on ITD. Reviewers agree the problem setting (non-i.i.d./temporal correlation) is important, but overall find the benchmark novelty, experimental coverage, and analysis depth insufficient for ICLR.  No rebuttal provided.

**Reviewer Concerns:**

No rebuttal is provided, so reviewer concerns remain outstanding: abTe questions novelty vs existing temporal/streaming TTA benchmarks, asks for reproducibility/split/release details, and flags weak justification for ADVMEM plus writing issues; AaSt raises limited realism (fixed-size crops, weak dynamics/occlusion), narrow coverage, synthetic corruptions, lack of multimodality, and missing newer temporal/memory TTA baselines; Hfyn argues novelty is unconvincing, baselines are outdated/incomplete, the ITD-trained base models are too weak to support conclusions, and ADVMEM lacks mechanistic analysis; urTJ also finds missing comparisons to closely related temporal-correlation setups and unclear articulation of what new failure modes ITD exposes.

**Reviewer Scores:**

No rebuttal provided, and thus likely no changes in reviewers scores.
- abTe: 2
- AaSt: 4
- Hfyn: 2
- urTJ: 4

---

### Decision · Program_Chairs · 2026-01-26

Reject